# DPNR: A DUAL-PROTOTYPE NOISE REPOSITORY BASED ON PROTOTYPE LEARNING FOR ANOMALY DETECTION

## ABSTRACT

Reconstruction-based methods have demonstrated remarkable success in the domain of industrial anomaly detection.Recently, Diffusion Models have been widely applied to industrial anomaly detection, driven by their powerful reconstruction capabilities. However, existing methods predominantly rely on an idealized Gaussian noise assumption. This creates a significant discrepancy with the complex and structured characteristics of anomalies in real-world industrial settings, leading to issues such as unpredictable model behavior and high false alarm rates. To address the aforementioned challenges, we are the first to introduce the concept of prototype learning into the domain of industrial anomaly detection. We formulate the Dual-Prototype Noise Repository (DPNR), a framework designed to guide the generation of realistic, structured noise and thereby replace the simplistic Gaussian noise prior. Specifically, DPNR guides a Dual-Prototype Guided Structured Noise Injection (DP-SNI) mechanism, enabling a dynamic and content-aware noise generation process. To address the limitations of traditional loss functions, we design the Region-adaptive Hybrid Noise Loss (RHN-Loss). By leveraging dynamic blending and adaptive weighting schemes, RHN-Loss provides robust and end-to-end optimizeable guidance.

## 1 INTRODUCTION

Anomaly detection has become a cornerstone technology in industrial manufacturing, enabling precise identification and localization of subtle anomalies in both product appearance and functional performance Chen et al. (2023); Liu et al. (2023); Liznerski et al. (2020).This capability is fundamental to maintaining stringent quality control standards while ensuring stable operational continuity across production lines.In practical industrial applications, anomalous samples are inherently rare and exhibit an imbalanced class distribution, whereas embedding-based anomaly detection methods Chen et al. (2024); Lee et al. (2022)require large-scale annotated anomaly data.Furthermore, product iterations and production line upgrades lead to the continuous emergence of new anomaly types.Consequently, supervised models, which are trained on a predefined library of known anomalies Gu et al. (2024); Jeong et al. (2023), has limitations in generalizing to these unknown anomalies. These critical drawbacks render the supervised learning paradigm unsuitable for the demands of real-world applications.Consequently, supervised reconstruction-based methods He et al. (2024); Cai et al. (2024)have gained widespread adoption in industrial inspection.These methods effectively address critical challenges such as high annotation costs and the dynamic evolution of anomalies.Against this backdrop, diffusion models, leveraging their unique noise learning mechanism, have opened up a new avenue for unsupervised anomaly detection Mousakhan et al. (2024); Zhang et al. (2024b).Diffusion models achieve unsupervised anomaly detection by reconstructing the features of normal samples and then utilizing the reconstruction error to identify anomalies Zhang et al. (2024b); Yao et al. (2024); Yin et al. (2023).The standard procedure involves two main stages: a forward diffusion process that progressively transforms the data distribution into standard Gaussian noise, and a subsequent reverse denoising process that generates the reconstructed image from this noise.A primary limitation of these methods is their reliance on Gaussian noise as the standard noise source. Gaussian noise lacks the topological structures and spatial correlations characteristic of real-world defects (e.g., scratches, cracks).Consequently, models trained

with such noise struggle to effectively locate or identify structured anomalies, revealing a significant discrepancy between the simplified noise assumption and the complexity of actual anomalies. Real-world noise, such as artifacts from electrical arcing on metal surfaces, is often impulsive and heavy-tailed. Unlike unstructured random pixels, these artifacts possess distinct morphology, texture, and spatial correlation, posing a significant challenge to conventional denoising methods. Noise in industrial defect datasets is characteristically diverse, structured, and non-Gaussian.

Consequently, models trained exclusively on Gaussian noise often confound normal surface textures with anomalous anomalies, resulting in a prohibitively high false positive rate. Training on a simplistic, unimodal Gaussian distribution is highly susceptible to overfitting, which severely compromises the model's ability to generalize to the diverse manifold of real-world anomalies. The intricate and non-Gaussian nature of industrial anomaly noise thus necessitates a departure from the conventional Gaussian assumption, calling for both the acquisition of structured noise and its integration into diffusion models. A critical challenge that inevitably arises from the shift from Gaussian to structured noise is the design of the loss function.

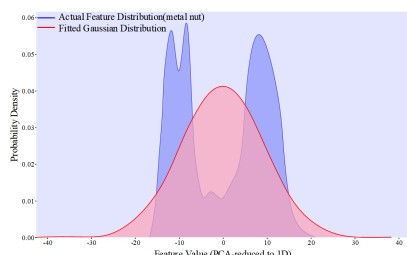

Figure 1: Feature Distribution of Real Noise vs. Idealized Gaussian Noise for the Metal Nut Class.

To bridge the gap between theoretical models and real-world applications, we depart from treating "noise" as a single, homogeneous distribution. Instead, we reframe it as a learnable and diverse ensemble. We are the first to introduce prototype learning to the field of industrial anomaly detection. We propose the Dual-Prototype Noise Repository (DPNR), a framework that learns and distills representative anomalous and normal prototypes from real-world industrial data. This repository is then used to guide a generative model in synthesizing high-fidelity, structured noise, effectively replacing the simplistic Gaussian noise assumption. Furthermore, for the model's noising process, we introduce a Dual-Prototype Guided Structured Noise Injection (DP-SNI) mechanism. By decoupling and learning separate representations for the background and foreground anomalies, DP-SNI synthesizes high-fidelity, structured noise that is endowed with spatial semantics. Through a learnable dual-noise parameterizer, we transform noise from a static, uninformative prior into a dynamic generative process that is conditioned on image content, has learnable parameters, and is semantically disentangled. Finally, both types of structured noise are coherently fused in a spatially-aware manner and injected into the training process of the diffusion model. This approach addresses the key limitations of diffusion models trained exclusively on Gaussian noise, namely their poor generalization to diverse real-world defects, susceptibility to overfitting, and unpredictable behavior. To address the limitations of conventional loss functions, we propose the Region-Adaptive Hybrid Noise Loss. This loss is specifically designed to resolve a core dilemma: the model must learn the global morphology of the complex, injected structured noise, while simultaneously and precisely predicting the ideal noise component essential for image reconstruction. We summarize our contributions as follows:

- The primary novelty of this work lies in being the first to conceptualize and apply prototype learning to the challenges of industrial defect detection. To this end, we construct a Dual-Prototype Noise Repository (DPNR), a framework based on prototype learning. DPNR learns the characteristics of noise distributions from real-world industrial data, thereby generating realistic, structured noise.

- Building upon an industrial data-driven approach, dual-prototype learning framework, we introduce the Dual-Prototype Guided Structured Noise Injection (DP-SNI) mechanism. This mechanism is specifically designed to inject authentic noise signals into the forward noising process of diffusion models.

- We formulate the Region-adaptive Hybrid Noise Loss (RHN-Loss) to overcome the limitations of traditional losses in guiding structured noise prediction. By leveraging dynamic blending, along with spatial and temporal weighting schemes, RHN-Loss balances noise

morphology reconstruction against the ideal denoising goal. This end-to-end optimizable loss significantly improves the model's capability to handle complex noise.

## 2 RELATED WORK

We briefly review related work about anomaly detection, diffusion models and prototype learning.

**Anomaly Detection:**The mainstream anomaly detection approaches can be primarily categorized into two paradigms: Reconstruction-based Methods and Embedding-based Methods.Embedding-based methods operate by first extracting feature embeddings from images using a pre-trained network. A normative model of the feature distribution is then established using only normal samples. Subsequently, anomalies are identified by measuring the deviation of a test feature from this normal model, which yields an anomaly score Wu et al. (2023); Bae et al. (2023); Gudovskiy et al. (2022); Zhou et al. (2023).

Reconstruction-based methods operate by localizing anomalous regions based on the discrepancy between an input image and its reconstruction.Classic early methods, such as autoencoders, leverage an encoder-decoder architecture to learn a low-dimensional manifold of normal data. Anomalies are subsequently identified based on high reconstruction error, as they deviate from this learned manifold Zhao et al. (2017); Liu et al. (2020); Bergmann et al. (2018).GAN-based methods Liang et al. (2023)detect anomalies by training a generator to model the distribution of normal data. For instance, AnoGAN Schlegl et al. (2019)leverages discrepancies in the discriminator's feature space for detection, while GANomaly Akcay et al. (2018)achieves this through reconstruction error in the latent space.DREAM Zavrtanik et al. (2021)proposes a collaborative generator-discriminator framework based on a UNet Ronneberger et al. (2015)architecture. Within this framework, the generator is responsible for reconstruction, while the discriminator is adversarially trained to identify anomalies within the reconstruction residuals.Driven by their powerful reconstruction capabilities, Diffusion Models have been increasingly applied to the field of industrial anomaly detection in recent years.

**Diffusion model:**Inspired by non-equilibrium thermodynamics, Diffusion Models are a class of generative models that learn a data distribution by reversing a gradual noising process. Their capacity for high-fidelity data generation and reconstruction enables their use in anomaly detection, where anomalous regions are localized by measuring the discrepancy between an input image and its reconstruction Zhang et al. (2023b); Tebbe & Tayyub (2024); Rombach et al. (2022).DiffusionAD employs standard-guided Gaussian mixture noise Zhang et al. (2023a).AnomalyDiffusion formulates few-shot conditioned Gaussian noise Hu et al. (2024).DiAD presents a hybrid strategy of class-conditioned Gaussian noise and frequency-domain adaptive noise Wyatt et al. (2022).All these current methods are built upon a "Gaussian noise prediction" training framework. However, the noise distribution in real-world industrial scenarios is often non-Gaussian.Since the noise added to each image varies, the reconstruction process becomes uncontrollable.Our formulated DPNR method replaces Gaussian noise with real structured noise, thereby overcoming the inherent limitations of the methods above and proving more effective for real-world industrial anomaly detection scenarios. **Prototype Learning:**Prototype learning is the process of learning a set of representative samples from large-scale data, which can effectively summarize the core characteristics of the data distribution.Owing to its high efficiency and inherent interpretability, prototype learning has been widely applied in various fields of computer vision.In the field of few-shot learning, some studies have proposed constructing a global prototype to capture generic visual knowledge that transcends single-class boundaries Liu et al. (2024).In segmentation tasks, researchers have proposed an adaptive prototype learning and assignment mechanism, which allows prototypes to adjust based on the image content dynamically Li et al. (2021).Recently, some works have begun to incorporate external knowledge, for instance, by leveraging semantic information to enhance the discriminative power of prototypes Zhang et al. (2024a).In semantic segmentation, PRCL combines prototype learning with contrastive learning Xie et al. (2024).PEM efficiently generates segmentation masks by using class prototypes as queriesCavagnero et al. (2024).

These studies demonstrate the significant role of prototypes in pixel-level understanding tasks. However, they share a common characteristic: prototypes are primarily used as a discriminative tool, with the ultimate goal of classification or matching, to determine which known prototype a pixel or region belongs to. This paper is the first to introduce prototype learning to the field of industrial anomaly

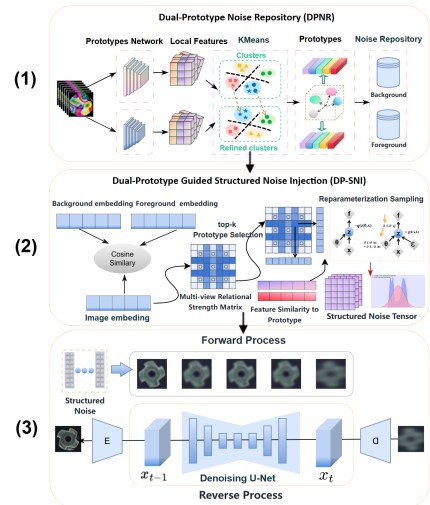

Figure 2: Overview of our three-stage framework. **(1)** A Dual-Prototype Noise Repository (DPNR) is built from representative prototypes distilled from real images. **(2)** The DP-SNI mechanism then uses these prototypes to dynamically generate content-aware structured noise. **(3)** This realistic structured noise, rather than Gaussian noise, is used to corrupt clean images, training the Denoising U-Net to learn robust, discriminative representations.

detection, and it elevates the application of prototypes to a new dimension: from a discriminative tool to a mechanism for generative guidance.By using disentangled normal and anomalous prototypes, we can generate specifically structured noise to replace traditional Gaussian noise, making the training process more targeted and controllable.

## 3 METHOD

Existing diffusion models, due to their reliance on Gaussian noise, are unable to model real-world industrial anomalies effectively. To address this issue, we are the first to introduce prototype learning to the field of industrial anomaly detection and formulate a comprehensive anomaly detection framework based on it. This framework guides the generation of realistic noise through a Dual-Prototype Noise Repository (DPNR). Based on the aforementioned prototype repository, we design a Structured Noise Injection mechanism (DP-SNI) that replaces the traditional process of injecting Gaussian noise. Meanwhile, to address the training difficulties introduced by structured noise, we propose a Region-wise Hybrid Noise Loss (RHN-Loss) function. This function effectively balances the dual objectives of noise prediction and image reconstruction, ensuring that the entire framework can be optimized end-to-end.

**Problem Definition**

First, we provide some background knowledge on diffusion models for subsequent analysis. In a standard diffusion model, the forward process progressively adds Gaussian noise to the original data $x_0$. After T steps, the noised sample $x_t$ can be directly obtained using the following equation:

$$x_t = \sqrt{\alpha_t}x_0 + \sqrt{1 - \alpha_t}\epsilon \tag{1}$$

$\alpha_t \in (0, 1)$ is a predefined noise scheduling coefficient that decreases as $t$ increases. However, in the reverse denoising process, the noise $\epsilon$ is unknown. A neural network $\epsilon_\theta(x_t, t)$ controlled by parameters $\theta$ is used to approximate this unknown true noise. By replacing $\epsilon$ with the model's prediction $\epsilon_\theta$, we obtain an estimate of $x_0$, which is denoted by $\hat{\mathbf{X}}_{t\to0}$. This demonstrates that predicting the noise $\epsilon_\theta$ is equivalent to directly predicting the initial $x_0$. The relationship is expressed by the following formula:

$$\mathbf{x}_t = \sqrt{\bar{\alpha}_t}\hat{\mathbf{x}}_{t\to0} + \sqrt{1 - \bar{\alpha}_t}\epsilon_\theta(\mathbf{x}_t, t) \tag{2}$$

The state at each step of the generation process is represented by $\hat{\mathbf{X}}_t$. The state after performing one denoising step is $\hat{\mathbf{X}}_{t-1}$. The process is expressed as follows:

$$\hat{\mathbf{x}}_{t-1} = \sqrt{\bar{\alpha}_{t-1}}\hat{\mathbf{x}}_{t\to0} + \sqrt{1 - \bar{\alpha}_{t-1}}\epsilon_\theta(\hat{\mathbf{x}}_t, t) \tag{3}$$

where $\hat{\mathbf{x}}_t$ is the intermediate result computed by the model from m of the previous step in the reverse denoising process. In real-world industrial scenarios, the noise is not unstructured random noise, but rather structured noise with complex spatial characteristics. Relying solely on a single type of Gaussian noise often leads to poor model controllability or the generation of unintended artifacts. To overcome the limitations above, we modify this process to handle real-world noise. Replace the Gaussian noise $\epsilon$ with a real structured noise sample $\eta$. The generalized forward equation is given by:

$$\mathbf{x}_t = \sqrt{\bar{\alpha}_t}\mathbf{x}_0 + \sqrt{1 - \bar{\alpha}_t}\eta \tag{4}$$

Correspondingly, the model's prediction target is shifted from Gaussian noise to the structured real noise, which is denoted as $\eta_\theta$. The original clean signal $x_t$ can be recovered from the image $x_0$, as follows:

$$\hat{\mathbf{x}}_{t\to0} = \frac{\mathbf{x}_t - \sqrt{1 - \bar{\alpha}_t}\eta_\theta(\mathbf{x}_t, t)}{\sqrt{\bar{\alpha}_t}} \tag{5}$$

where the model's prediction $\eta_\theta(\mathbf{x}_t, t)$ is substituted for the unknown real noise $\eta$. By predicting and subtracting the structured noise component $\eta_\theta$, the clean image $\hat{\mathbf{x}}_{t\to0}$ is isolated.

**Dual-Prototype Noise Repository**

To enable our model to learn noise with complex spatial structures found in real-world scenarios, we construct a Dual-Prototype Structured Noise Memory. This memory serves as a compact, non-redundant repository to provide diverse structured noise for downstream tasks. We partition the original dataset of noise image patches $D$, into two mutually exclusive subsets: the foreground noise set $D_{fg}$, the background noise set $D_{bg}$, satisfying $D = D_{fg} \cup D_{bg}$ and $D_{fg} \cap D_{bg} = \emptyset$. The objective is to construct a compact prototype memory for each of these two subsets, denoted as $\mathcal{M}_{fg}$ and $\mathcal{M}_{bg}$.

We begin by embedding each raw noise patch $p \in D$ into a high-dimensional latent space. The features are extracted using the extractor $\Phi : \mathbb{R}^{H \times W \times C} \to \mathbb{R}^D$. For each noise patch $p_i$ in the set, we compute its corresponding feature vector $\mathbf{f}_i$ as follows:

$$\mathbf{f}_i = \Phi(p_i) \quad \text{where} \quad \mathbf{f}_i \in \mathbb{R}^D \tag{6}$$

This process yields the feature set $\mathcal{F} = \{\mathbf{f}_1, \mathbf{f}_2, \ldots, \mathbf{f}_N\}$, where $N = |\mathcal{D}|$. Directly operating on the vast feature set $\mathcal{F}$ is inefficient. Our goal is to identify a set of K representative points (i.e., initial prototypes) that summarize the entire feature space. We employ the K-Means clustering algorithm to achieve this objective. The K-Means algorithm partitions the feature set $\mathcal{F}$ into $K$ clusters, $S = \{S_1, S_2, \ldots, S_K\}$, by finding the centroid $\mathbf{c}_k$ for each cluster. These centroids, which serve as our initial prototypes, are optimized to minimize the within-cluster sum of squares, as formulated below:

$$\arg\min_{S,\{\mathbf{c}_k\}_{k=1}^K} \sum_{k=1}^{K} \sum_{\mathbf{f}_i \in S_k} \|\mathbf{f}_i - \mathbf{c}_k\|_2^2 \tag{7}$$

where $\mathbf{c}_k = \frac{1}{|S_k|}\sum_{\mathbf{f}_i \in S_k} \mathbf{f}_i$ is the centroid of the cluster $\mathcal{S}_k$. The resulting set of centroids $\mathbf{C}_{initial} = \{\mathbf{c}_1, \mathbf{c}_2, \ldots, \mathbf{c}_K\}$, constitutes the initial prototype memory. The initial prototype set $\mathbf{C}_{initial}$, may contain semantic redundancy, meaning that multiple prototypes could represent highly similar noise patterns. To perform redundancy removal, we measure the semantic similarity between any two initial prototypes $\mathbf{c}_i$ and $\mathbf{c}_j$, using cosine similarity.

$$S(\mathbf{c}_i, \mathbf{c}_j) = \frac{\mathbf{c}_i^\top \mathbf{c}_j}{\|\mathbf{c}_i\|_2 \cdot \|\mathbf{c}_j\|_2} \tag{8}$$

We define a similarity relation, denoted by $\sim$. We say $\mathbf{c}_i \sim \mathbf{c}_j$ that if $S(\mathbf{c}_i, \mathbf{c}_j) > \tau$, where $\tau$ is a predefined high-similarity threshold.

$$\eta_l = \frac{1}{|G_l|} \sum_{\mathbf{c}_j \in G_l} \mathbf{c}_j \tag{9}$$

Based on this relation, we partition the initial prototype set $\mathbf{C}_{initial}$ into $M$ mutually exclusive equivalence classes: $G_1, G_2, \ldots, G_M$. For each equivalence class $G_l$, a single, refined prototype $\eta_l$ is generated by computing the mean of all the initial prototypes it contains. Through this process, the initial $K$ prototypes are refined into $M$ final prototypes (where $M \leq K$), which constitute the final prototype memory $\mathcal{M} = \{\eta_1, \eta_2, \ldots, \eta_M\}$.

**Dual-Prototype Guided Structured Noise Injection**

To overcome the performance bottleneck of diffusion models in anomaly detection, which stems from their reliance on unstructured Gaussian noise, we propose a dual-prototype-guided structured noise injection mechanism. This mechanism shifts the noise generation process from a fixed, data-agnostic prior to a learnable, dynamic process that is guided by the input image content. The process begins by projecting a normal image $\mathbf{x}_s$ and an anomalous image $\mathbf{x}_a$ into a latent space via a pre-trained encoder $\epsilon$. This projection results in the latent vectors $\mathbf{z}_s$ and $\mathbf{z}_a$ respectively.

$$\mathbf{z}_s = c_{\text{scale}} \cdot \mathcal{E}(X_s) \tag{10}$$

$$\mathbf{z}_a = c_{\text{scale}} \cdot \mathcal{E}(X_a) \tag{11}$$

The scaling factor is denoted by $c_{\text{scale}}$.

We dynamically bridge the normal and anomalous representations by performing a linear interpolation between their latent vectors. The interpolation ratio is controlled by the timestep $t \in [0, T-1]$, resulting in a synthetic vector $\mathbf{z}_{\text{syn}}(t)$ that smoothly evolves from normal to anomalous.

$$\mathbf{z}_{\text{syn}}(t) = (1 - \alpha_t)\mathbf{z}_s + \alpha_t \mathbf{z}_a \quad \text{where} \quad \alpha_t = t/T \tag{12}$$

This vector $\mathbf{z}_{\text{syn}}(t)$ is then fed into the diffusion process, where the model is trained on the self-reconstruction task of restoring the original $\mathbf{z}_{\text{syn}}(t)$ from its noisy variants. Our data-driven dual-prototype representation learning component is designed to disentangle the input normal image $\mathbf{x}_s$ into distinct background and foreground feature representations, which in turn serve as guidance signals. We first preprocess the normal image and then pass it through the feature extractor $\Phi$ to obtain its hierarchical feature representations. These multi-level features are then integrated through weighted fusion to yield a single, semantically rich, dense feature map, denoted as $\mathbf{F} \in \mathbb{R}^{N \times D}$.

$$\mathbf{F} = \sum_{l \in \mathcal{L}} w_l \cdot \Phi^{(l)}(\mathbf{x}_s) \tag{13}$$

where $\mathcal{L}$ is the set of selected layers, $w_l$ are the fusion weights for each layer, $N$ is the number of image patches, $D$ is the feature dimension.

The prototype matching and feature disentanglement step introduces two sets of learnable prototypes: a set of background prototypes $\mathcal{P}_{bg}$ and a set of foreground prototypes $\mathcal{P}_{fg}$. We design a Prototype Matcher, denoted as $\mathcal{H}$, which calculates the relevance between each image patch feature and the two prototype sets via an attention mechanism. It outputs three key components: (1) a disentangled global background feature $\mathbf{f}_{bg}$, (2) a disentangled global foreground feature $\mathbf{f}_{fg}^{\alpha}$, and (3) a foreground attention map $\mathbf{A}_{fg} \in \mathbb{R}^N$, which represents the confidence score of each patch belonging to the foreground.

$$(\mathbf{f}_{bg}, \mathbf{f}_{fg}, \mathbf{A}_{bg}, \mathbf{A}_{fg}) = \mathcal{H}(\mathbf{F}, P_{bg}, P_{fg}) \tag{14}$$

The primary objective of the "Learnable Structured Noise Generation and Mixing" stage is to transform the abstract feature vectors $\mathbf{f}_{bg}$ and $\mathbf{f}_{fg}$, obtained from the previous step, into a concrete, injectable structured noise $\boldsymbol{\varepsilon}_s$. We design two independent Noise Parameterizers, denoted as $\boldsymbol{\Psi}_{bg}$ and $\boldsymbol{\Psi}_{fg}$. They map the disentangled feature vectors to the statistical parameters of a Gaussian distribution: the mean $\mu$ and the log-variance $\log \sigma^2$.

$$(\boldsymbol{\mu}_{bg}, \log \boldsymbol{\sigma}_{bg}^2) = \Psi_{bg}(\mathbf{f}_{bg}) \tag{15}$$

$$(\boldsymbol{\mu}_{fg}, \log \boldsymbol{\sigma}_{fg}^2) = \Psi_{fg}(\mathbf{f}_{fg}) \tag{16}$$

We then generate two noise bases $\boldsymbol{\varepsilon}_{\text{bg\_base}}$ and $\boldsymbol{\varepsilon}_{\text{fg\_base}}$, by sampling from these two learned distributions using the reparameterization trick. The resulting samples are passed through a tanh activation function to enhance their structural properties and constrain their value range.

$$\boldsymbol{\sigma} = \exp(0.5 \cdot \log \boldsymbol{\sigma}^2) \tag{17}$$

$$\mathbf{z} \sim \mathcal{N}(\mathbf{0}, \mathbf{I}) \tag{18}$$

$$\varepsilon_{\text{base}} = C \cdot \tanh(\boldsymbol{\mu} + \mathbf{z} \odot \boldsymbol{\sigma}) \tag{19}$$

where $C$ is a scaling constant and $\odot$ denotes element-wise multiplication.

To fuse the two noise bases, we propose a Spatially-Aware Noise Blending method. First, the foreground attention map $\mathbf{A}_{fg}$ to generate a foreground mask $\mathbf{M}_{fg}$ at the latent space resolution. We then apply Gaussian smoothing to this mask. This step is crucial for modeling the gradual, natural-looking transitions between foreground anomalies and the background, as detailed in the formula below:

$$\mathbf{M}_{\text{smooth}} = \mathcal{G}_{\text{blur}}(\mathbf{M}_{fg}) \tag{20}$$

Simultaneously, the low-dimensional noise bases $\varepsilon_{\text{bg\_base}}$ and $\varepsilon_{\text{fg\_base}}$, are upsampled to the latent space dimensions using bicubic interpolation $\mathcal{U}^{\text{bicubic}}$. The final structured noise $\mathcal{E}_s$, is obtained through a spatial weighted blend using the smoothed mask:

$$\varepsilon_s = (1 - \mathbf{M}_{\text{smooth}}) \odot \mathcal{U}_{\text{bicubic}}(\varepsilon_{\text{bg\_base}}) + w_{fg} \cdot \mathbf{M}_{\text{smooth}} \odot \mathcal{U}_{\text{bicubic}}(\varepsilon_{\text{fg\_base}}) \tag{21}$$

where $w_{fg}$ is a hyperparameter that controls the intensity of the foreground noise. Finally, we inject the generated structured noise $\mathcal{E}_s$, into the synthetic latent feature $\mathbf{z}_{\text{syn}}(t)$ to simulate the forward noising step of the diffusion process.

**Region-adaptive Hybrid Noise Loss**

To effectively guide the diffusion model's learning process under our structured noise injection mechanism, we designed the Region-Adaptive Mixed-Noise Loss function. The model must not only learn the injected structured noise $\varepsilon_s$, but also accurately predict the ideal noise component $\varepsilon$, which is essential for image inpainting. We decouple the model's learning task into two parallel distribution-matching objectives. The first, the $\varepsilon$-prediction objective, follows the standard training paradigm of diffusion models, aiming for the accurate reconstruction of the original noise-free image $\mathbf{x}_0$. This is equivalent to minimizing the Mean Squared Error (MSE) between the model's prediction $\boldsymbol{\epsilon}_\theta(\mathbf{x}_t, t)$ and the ground-truth sampled noise $\boldsymbol{\epsilon}$.

$$\mathcal{L}_{\text{eps}} = \mathbb{E}_{\mathbf{x}_0, \boldsymbol{\epsilon}, t} \left[ \|\boldsymbol{\epsilon} - \boldsymbol{\epsilon}_\theta(\mathbf{x}_t, t)\|^2 \right] \tag{22}$$

where $\mathcal{L}_{\text{eps}}$ represents the ideal denoising objective.

The structured noise matching objective is designed to compel the model to learn and recognize the structured noise $\boldsymbol{\epsilon}_s$. We model this problem as minimizing the Kullback-Leibler (KL) divergence between the two Gaussian distributions defined by the model's prediction $\varepsilon_s$ and the target noise $\boldsymbol{\epsilon}_s$. Under simplifying assumptions, this is equivalent to:

$$\mathcal{L}_{\text{struct}} = \mathbb{E}_{\boldsymbol{\epsilon}_s, t} \left[ \frac{\|\boldsymbol{\epsilon}_\theta(\mathbf{x}_t, t) - \boldsymbol{\epsilon}_s\|^2}{2 \cdot \text{Var}(\boldsymbol{\epsilon}_s)} \right] \tag{23}$$

where $\text{Var}(\boldsymbol{\epsilon}_s)$ calculates the variance of the target structured noise $\boldsymbol{\epsilon}_s$. $\mathcal{L}_{\text{struct}}$ represents the structured noise reconstruction loss.

To reconcile these two inherently conflicting objectives, we design an adaptive mechanism. The first component is a learnable dynamic blending strategy, which is realized by introducing a learnable parameter $\forall \alpha \in \mathbb{R}$. It is mapped to the (0,1) interval via the Sigmoid function $\sigma(\cdot)$ to serve as the convex combination weight for two losses, forming the mixed loss $\mathcal{L}_{\text{hybrid}}$:

$$\mathcal{L}_{\text{hybrid}} = \sigma(\alpha) \cdot \mathcal{L}_{\text{struct}} + (1 - \sigma(\alpha)) \cdot \mathcal{L}_{\text{eps}} \tag{24}$$

where $\alpha$ is a hyperparameter that controls the weight of the background region.

Finally, we re-weight the loss according to the diffusion timestep t. This is motivated by the fact that denoising steps at different stages contribute differently to the final generation quality. We employ a re-weighting scheme based on the Signal-to-Noise Ratio (SNR) for each timestep. The noise scheduler defines the cumulative variance $\bar{\alpha}_t$. For a given timestep t, the Signal-to-Noise Ratio (SNR) can then be defined as $\text{SNR}(t) = \frac{\bar{\alpha}_t}{1 - \bar{\alpha}_t}$. Our weighting function $w(t)$, is set to be the Signal-to-Noise Ratio (SNR). In our method, we adopt a cosine schedule for $\bar{\alpha}_t$, and consequently, the

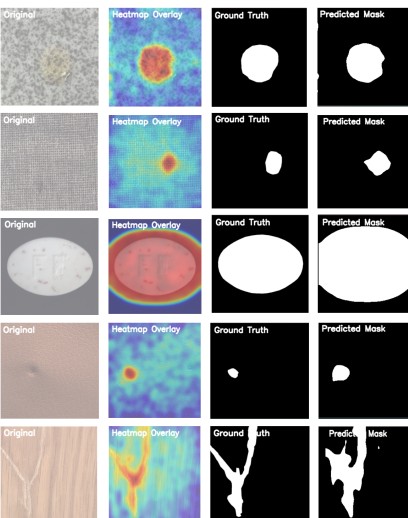

Figure 3: Qualitative results of our proposed method on several categories from the MVTec AD dataset. From top to bottom, the categories shown are: tile, grid, pill, leather, and wood. For each sample, we display the original image, the anomaly heatmap, the ground-truth defect mask, and the final predicted mask generated by our model. The results demonstrate that our model can effectively localize and segment various types of anomaly.

weighting function is formulated as:

$$\bar{\alpha}_t = \cos^2\left(\frac{\pi t}{2T}\right) \tag{25}$$

$$w(t) = \frac{\bar{\alpha}_t}{1 - \bar{\alpha}_t} \tag{26}$$

Multiplying the loss by $w(t)$ effectively increases the weight for timesteps with a low Signal-to-Noise Ratio (SNR), a practice that has been shown to help stabilize training and improve generation quality.

By combining all the above components, we arrive at the final, complete loss function expression computed in each training step. This expression integrates the dual objectives, learnable mixing, spatial weighting, and timestep weighting to form a comprehensive optimization objective:

$$\mathcal{L}_{\text{total}} = \mathbb{E}_{x_0,\epsilon_r,M,t}\left[\text{clamp}(w(t)) \cdot \text{mean}(\mathcal{L}_{\text{spatial}})\right] \tag{27}$$

The expanded form is:

$$L_{\text{total}} = \mathbb{E}_{p(x_0,\epsilon,M,t)}\left[\text{clamp}\left(\frac{\alpha_t}{1 - \alpha_t}\right)\right.$$

$$\cdot \text{mean}\left((\sigma(\alpha)L_{\text{struct}} + (1 - \sigma(\alpha))L_{\text{eps}})\right. \tag{28}$$

$$\left.\left.\odot (\sigma(\beta)M + c(1 - M)))\right)\right]$$

The *clamp* operation is used to limit the maximum value of the timestep weight, which increases training stability. This loss function is key to the successful implementation of the entire framework.

## 4 EXPERIMENTS

**Comparison with SOTAs**

| Category | Non-Diffusion Method | | | | Diffusion-based Method | | |
|---|---|---|---|---|---|---|---|
| | PaDiM | MKD | DRAEM | UniAD | DDPM | LDM | Ours |
| Bottle | 96.1/- | 91.8/- | 87.6/62.95/56.9/- | 98.1/66.0/69.2/- | 59.9/4.9/11.7/43.4 | 86.9/49.1/50.0/- | **89.2/52.2/52.9**/80.0 |
| Cable | 81.0/- | 89.3/- | 71.3/14.7/17.8/- | 97.3/39.9/45.2/- | 66.5/ 6.7/10.6/19.5 | 89.3/18.5/26.2/- | 65.0/ 7.7/14.3/28.1 |
| Capsule | 96.9/- | 88.3/- | 50.5/ 6.0/10.0/- | 98.5/43.4/50.0/- | 63.1/ 6.2/ 9.7/ 9.3 | 90.0/ 7.9/27.3/- | 72.0/ 7.5/19.0/21.2 |
| Hazelnut | 96.3/- | 91.2/- | 96.9/70.0/60.5/- | 98.1/55.2/56.8/- | 91.2/24.1/28.3/21.0 | 95.1/51.2/53.5/- | 90.4/12.6/18.2/75.6 |
| Metal Nut | 84.4/- | 64.2/- | 62.2/31.1/21.0/ | 94.8/55.5/66.4/- | 62.7/14.6/29.2/16.5 | 70.5/19.3/30.7/- | **78.7**/ 4.1/10.1/52.7 |
| Screw | 94.1/- | 92.1/- | 95.5/33.8/40.6/- | 98.3/28.7/37.6/- | 91.1/ 1.8/ 3.8/16.0 | 91.7/ 2.2/ 4.6/- | 79.6/ **7.6/13.6**/45.1 |
| Toothbrush | 95.6/- | 88.9/- | 97.7/55.2/55.8/- | 98.4/34.9/45.7/- | 76.9/ 4.0/ 7.7/16.0 | 93.7/20.4/ 9.8/- | **92.1**/ 8.3/ 3.9/71.5 |
| Transistor | 92.3/- | 71.7/- | 64.5/23.6/15.1/- | 97.9/59.5/64.6/- | 53.1/5.8/11.4/23.3 | 85.5/25.0/30.7/- | **91.1**/ 8.9/19.1/49.6 |
| Zipper | 94.8/- | 86.1/- | 98.3/74.3/69.3/- | 96.8/40.1/49.9/- | 67.4/ 3.5/ 7.6/36.7 | 66.9/ 5.3/ 7.4/- | **71.2/12.4/15.9**/25.8 |
| Carpet | 97.6/- | 95.5/- | 98.6/78.7/73.1/- | 98.5/49.9/51.1/- | 89.2/18.8/44.3/20.1 | 99.1/70.6/66.0/- | 93.2/54.2/57.3/84.3 |
| Grid | 71.0/- | 82.3/- | 98.7/44.5/46.2/- | 96.5/23.0/28.4/- | 63.1/ 0.7/ 1.9/12.3 | 52.4/ 1.1/ 1.9/- | **93.2/53.6/57.8**/84.3 |
| Leather | 84.8/- | 96.7/- | 97.3/60.3/57.4/- | 98.8/32.9/34.4/- | 97.3/38.9/43.2/25.9 | 99.0/45.9/44.0/- | 81.7/7.15/14.8/52.5 |
| Tile | 80.5/- | 85.3/- | 98.0/93.6/86.0/- | 91.8/42.1/50.6/- | 87.0/35.2/36.6/21.4 | 90.1/43.8/51.2/- | **99.2**/48.7/51.3/95.6 |
| Wood | 89.1/- | 80.5/- | 96.0/81.4/74/6/ | 93.2/37.2/41.5/- | 84.7/30.9/37.3/16.0 | 92.3/44.1/46.6/- | **96.5**/78.4/74.9/93.8 |

Table 1: Comparison with SOTA methods on MVTec-AD dataset for multi-class anomaly localization with AUROC/AP/F1max metrics.

For our comparative analysis, we categorize the competing methods into two main types: non-diffusion and diffusion-based methods. We then evaluate performance against a suite of models, including DRAEM, MKD, UniAD, DDPM, and LDM, across the MVTec-AD, VisA, and MPDD datasets, reporting results for every category.

**Ablation Studies**

The architectural design of DPNR is distinct from current methods. While existing approaches commonly add synthetic Gaussian noise, our model is engineered to incorporate real-world noise. Accordingly, we have designed a novel noise injection mechanism and a corresponding loss function for the model. Our ablation study demonstrates the effectiveness of our proposed components. When we bypass the noise injection mechanism and conduct experiments directly with unprocessed noise, the model's reconstruction capability is impaired. Similarly, when our designed loss function module is replaced by the standard loss used in conventional diffusion models, the model's performance degrades significantly. Furthermore, when we use Gaussian noise for the noising process while still employing our designed loss function, the model's performance also declines. This indicates that our loss function is specifically optimized for the characteristics of real-world noise. These experiments demonstrate that when applying real-world noise to diffusion models, a corresponding loss function is indispensable for achieving performance improvements. This validates the effectiveness of our proposed method.

| DPNR | DP-SNI | RHN-Loss | Tile | Wood | Grid |
|---|---|---|---|---|---|
| ✓ | | | 67.3 | 75.6 | 70.46 |
| ✓ | ✓ | | 82.3 | 89.3 | 79.3 |
| ✓ | ✓ | ✓ | 99.2 | 96.5 | 93.2 |

Table 2: Ablation studies on the design of DPNR with AUROC metrics.

## 5 CONCLUSION

This paper proposed DPNR addresses a core limitation in current diffusion-based methods for industrial anomaly detection: their fundamental reliance on Gaussian noise, which leads to unpredictable model behavior and high false positive rates. Pioneering the use of prototype learning in industrial anomaly detection, we introduce the DP-SNI mechanism to dynamically inject structured noise. This novel approach is complemented by our specifically designed RHN-Loss function, creating a cohesive and effective framework. Extensive experiments on the MVTec-AD, MPDD, and VisA benchmark datasets have empirically validated the effectiveness of our proposed DPNR framework. As this work represents a pioneering effort in replacing the traditional Gaussian noise with a structured noise learning paradigm, we acknowledge that there is room for enhancement in handling certain anomaly categories. In our future work, we will focus on further exploration to improve the quality of the noise repository and to design more fine-grained processing methods for structured noise, ultimately aiming to achieve higher reconstruction performance.

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
