# OpenReview forum: "DPNR: ADual-Prototype Noise Repository based on Prototype Learning for  Anomaly Detection"
_ICLR.cc/2026/Conference — Submitted to ICLR 2026_

### Official Review · Reviewer_mNS5 · 2025-10-25

**Soundness:** 2
**Presentation:** 1
**Contribution:** 2
**Rating:** 2
**Confidence:** 4

**Summary:**

This paper presents DPNR, a dual-prototype noise repository for diffusion-based anomaly detection. The authors argue that standard Gaussian noise fails to capture the structured, spatially correlated nature of real industrial defects. DPNR introduces (1) a dual-prototype repository that clusters normal and abnormal noise patterns, and (2) a region-adaptive hybrid noise loss (RHN-Loss) combining reconstruction and denoising objectives.

**Strengths:**

- The motivation to improve the realism of diffusion-based noise modeling is intuitive and relevant to industrial anomaly detection.
- The proposed prototype-guided noise design is conceptually interesting, offering a different perspective from Gaussian diffusion priors.

**Weaknesses:**

1. The paper contains several factual inaccuracies. For instance, the authors claim to be “the first to introduce prototype learning to anomaly detection,” which is clearly inaccurate given prior works such as ProtoAD and ,.etc has used the same concept. Such unsupported claims undermine the paper’s credibility and suggest a lack of proper literature review.
2. Although the paper has claimed multiple times of “validated on MVTec, VisA, and MPDD,” the main text only provides results for MVTec-AD. The other datasets are neither described nor reported, which is severe misleading.
3. The entire experiment section lacks clarity and rigor. Tables do not contains average metrics, no highlight of the best results, some baselines outperform the proposed method, and there is zero analysis or discussion explaining the results. This makes the empirical validation appear superficial and unconvincing.

**Questions:**

See weakness.

---

### Official Review · Reviewer_kWBF · 2025-10-31

**Soundness:** 2
**Presentation:** 2
**Contribution:** 2
**Rating:** 2
**Confidence:** 3

**Summary:**

This paper proposes an industrial anomaly detection framework, DPNR (Dual-Prototype Noise Repository), based on diffusion models, aiming to replace the traditional Gaussian noise prior. The authors introduce a prototype learning mechanism, which learns dual sets of normal and abnormal prototypes to generate structured noise, simulating more realistic industrial defect patterns.

**Strengths:**

Introducing prototype learning into the diffusion noise modeling space is conceptually innovative.

**Weaknesses:**

Weak experimental performance — Although conceptually interesting, the overall performance is low (Pixel AUROC ≈ 85.5), far below current diffusion or Transformer-based methods, limiting practical significance.
Lack of comparison with recent methods — The paper does not include comparisons with recent diffusion or foundation models (2024–2025).

**Questions:**

Missing experimental details — The paper claims that experiments were conducted on the VisA and MPDD datasets, yet no corresponding quantitative results or visual analyses are presented in the main text. Please clarify where these results can be found  or include them for completeness.

---

### Official Review · Reviewer_EpNF · 2025-10-31

**Soundness:** 2
**Presentation:** 1
**Contribution:** 2
**Rating:** 2
**Confidence:** 5

**Summary:**

DPNR proposes an innovative prototype learning-based approach that builds a Dual-Prototype Noise Repository to capture complex noise patterns in real industrial scenarios, replacing the Gaussian noise assumption in traditional diffusion models. By integrating a structured noise injection mechanism (DP-SNI) and a region-adaptive hybrid noise loss (RHN-Loss), it aims to improve anomaly detection accuracy and generalization for more reliable industrial applications. However, its experimental results and validation remain limited.

**Strengths:**

1.The paper innovatively introduces prototype learning into industrial anomaly detection through a Dual-Prototype Noise Repository that replaces Gaussian noise assumptions with learned normal and abnormal prototypes.

2.The proposed DP-SNI mechanism decouples noise generation into foreground and background branches with spatial-aware fusion, enabling more realistic defect-structured noise.

3.The RHN-Loss employs a dynamic weighting strategy to balance denoising and structured noise reconstruction objectives, improving training stability and adaptability.

**Weaknesses:**

1.The motivation and problem definition are inconsistent, as the method claims to address non-Gaussian noise but still relies on Gaussian-based reparameterization.

2.The experimental evaluation lacks sufficient ablation studies and detailed comparisons with recent diffusion-based anomaly detection methods.

3.Important implementation details such as prototype number, similarity threshold, feature extractor design, and hyperparameters are missing, reducing reproducibility.

4.The theoretical contribution is limited, as the method mainly combines existing ideas without demonstrating clear conceptual novelty.

**Questions:**

1.The necessity of prototype learning is unclear, and the distinction between the proposed dual-prototype design and traditional memory-bank approaches needs stronger justification.

2.The method’s complexity appears disproportionate to its marginal performance gains, raising questions about practical efficiency.

3.The assumption of real-world non-Gaussian noise may be conceptually inaccurate, as benchmark datasets primarily involve structural defects rather than true noise.

---

### Official Review · Reviewer_kxpN · 2025-11-01

**Soundness:** 2
**Presentation:** 2
**Contribution:** 3
**Rating:** 2
**Confidence:** 4

**Summary:**

This paper proposes a diffusion model-based framework for industrial image anomaly generation named DPNR, which introduces the concept of dual-prototype learning to replace traditional Gaussian noise injection. Additionally, the paper designs the Region-adaptive Hybrid Noise Loss (RHN-Loss) to optimize model training and validates the framework through experiments on multiple datasets.

**Strengths:**

1. Introduces prototype learning to generate dynamic, content-aware noise, which is an innovative idea with reasonable motivation.
2. The proposed RHN-Loss attempts to balance image reconstruction and noise modeling, offering some practical insight.

**Weaknesses:**

1. The paper does not logically explain why introducing prototype learning can generate more realistic and structured noise than Gaussian noise.
2. There is inconsistency between text and figures. Figure 2 does not fully correspond to Section “3 METHOD”. For example, Figure 2(2) actually includes part of the “Dual-Prototype Noise Repository” module.
3. The experimental results are insufficient: (1) No comparison with more recent methods from 2025; (2) Only partial pixel-level metrics for anomaly localization are provided (AUPRO should be added), and image-level metrics (I-AUROC) are missing; (3) The results on the VisA and MPDD datasets mentioned in the paper are not included; (4) There is a lack of detailed parameter ablation studies and discussion; (5) The module ablation experiments only include three categories of the MVTec-AD dataset, without average pixel-level and image-level results across all categories.
4. The visualization of detection results is limited. It is recommended to include additional qualitative results across multiple datasets and categories, and compare them with existing strong methods mentioned in Table 1.
5. The figures and tables contain excessive white space, the layout and size of Figures 2, 3, and Table 2 should be adjusted.

**Questions:**

1. The paper does not discuss the impact of the original dataset of noise image used for noise generation. Is DPNR robust to different types of noise datasets?
2. The authors divide the initial prototype set into M mutually exclusive equivalence classes, but there is no discussion of how M is chosen. Would an larger or smaller M affect the quality of generated anomaly samples?
3. From a logical perspective, why does introducing prototype learning lead to more realistic and structured noise compared to Gaussian noise?

---

### Meta-Review · Area_Chair_1Z5E · 2025-12-05

**Summary:**

All reviewers give a rate of 2, and authors do not engage in the discussion. The paper does not logically explain why introducing prototype learning can generate more realistic and structured noise than Gaussian noise. I carefully check the paper to confirm reviewers comment, thereby the paper should be rejected.

**Reviewer Concerns:**

The motivation is not clearly claimed, which does not logically explain why introducing prototype learning can generate more realistic and structured noise than Gaussian noise. Experiments are also insufficient.

**Reviewer Scores:**

Reviewers might keep their scores, since authors do not engage in the discussion.

---

### Decision · Program_Chairs · 2026-01-26

Reject